# The Association between Metabolic Syndrome and Biochemical Markers in Beijing Adolescents

**DOI:** 10.3390/ijerph16224557

**Published:** 2019-11-18

**Authors:** Yao Zhao, Yingjie Yu, Hong Li, Mingying Li, Dongran Zhang, Dandan Guo, Xiaohui Yu, Ce Lu, Hui Wang

**Affiliations:** 1Department of Nutrition and Food Hygiene, Beijing Center for Disease Prevention and Control, Beijing100013, China; zhaoy@bjcdc.org (Y.Z.); yuyingjie@bjcdc.org (Y.Y.); lihong@bjcdc.org (H.L.); guodandan@bjcdc.org (D.G.); yuxiaohui@bjcdc.org (X.Y.); 2Beijing Research Center for Preventive Medicine, Beijing 100013, China; 3Department of Nutrition and Food Hygiene, Center for Disease Control and Prevention of Xicheng District, Beijing 100013, China; limingying@bjxccdc.cn; 4Department of Nutrition and Food Hygiene, Center for Disease Control and Prevention of Fangshan District, Beijing 100013, China; fsspk427@163.com; 5Department of Epidemiology and Biostatistics, School of Public Health, Nanjing Medical University, No.101 Longmian Ave, Jiangning District, Nanjing 211166, China; luce@njmu.edu.cn

**Keywords:** MetS, ALT, SUA, CRP, LDL, adolescent

## Abstract

**Objective:** To describe the prevalence of metabolic syndrome (MetS) in adolescents and its association with several MetS-related biochemical markers. **Methods:** A cross-sectional analysis was carried out and data were extracted from the Nutrition and Health Surveillance in Primary and Secondary school students of Beijing (NHSPSB) 2017. Participants were aged 10–15 years old. MetS was diagnosed using the recommended criteria for Chinese adolescents. The associations among MetS, biochemical biomarkers, and socioeconomic status were estimated by multivariable linear regression. **Results:** The prevalence of MetS in adolescents in Beijing was 3% in the total sample, 4% in boys, and 2% in girls. Moreover, the prevalence of MetS in the overweight and obesity populations were 5% and 12% respectively. The prevalence of MetS remained higher in boys than in girls. The concentrations of alanine aminotransferase (ALT), serum uric acid (SUA), low density lipoprotein (LDL), and C-reactive protein (CRP) were higher in the MetS children in comparison with non-MetS children (All p < 0.05), while the high-density lipoprotein (HDL) concentration was lower in MetS children. After adjusting for socioeconomic parameters in the multivariable regression model, MetS was strongly associated with ALT, SUA, HDL, and LDL. The five components of MetS indicated that abdominal obesity and a high serum triglyceride (TG) concentration were tightly linked with ALT, SUA, LDL, and CRP; while a low HDL concentration and elevated blood pressure were related to enhanced ALT, UA, and CRP. Additionally, impaired fasting glucose was only related to increased ALT. **Conclusion:** The epidemiological issues of MetS in Beijing adolescents should be known across socioeconomic classes. Early intervention strategies, such as dietary pattern interventions and physical excise, should be designed for that population to reduce the disease burdens of cardiovascular disease (CVD), Type 2 diabetes (T2D), and steatohepatitis in adulthood.

## 1. Introduction

Metabolic syndrome (MetS) is a cluster of several interrelated abnormalities including abdominal obesity, elevated blood pressure, impaired fasting glucose levels, high values of triglycerides (TG), and low values of high-density lipoprotein (HDL) [1].The prevalence of MetS has been increasing dramatically around the world in the past decades. The age-standardized prevalence of MetS was 9.8% for men and 17.8% for women in Chinese adults in 2001 [2]. MetS in children has attracted increased attention recently, as it tends to be an important predictor of adulthood cardiovascular disease. He Yuna et al. found that the prevalence of MetS was 2.4% in Chinese adolescents (aged 10–17 years old) in 2012 [3]. MetS is considered a risk factor of cardiovascular disease, type 2 diabetes, and all-cause mortality [4]. In addition, other abnormalities, such as fibrinolysis, thrombosis, and inflammation are strongly associated with MetS [5].

Previous studies have indicated that MetS is associated with serum uric acid (SUA), C-reactive protein (CRP), and nonalcoholic fatty liver disease (NAFLD) in adults and adolescents. Uric acid is the end product of endogenous purine and dietary factors. The SUA level is impacted by diets rich in meat or seafood and varies across physical activity levels. Moreover, the SUA level is affected by the degree of insulin resistance and hypertension [6,7]. Higher insulin resistance or hypertension is associated with a higher concentration of SUA [6,7]. Additionally, the SUA level is positively related to the triglyceride concentration, which can be attributed to a reduction in the glomerular filtration rate caused by dyslipidemia [8,9,10]. The CRP level is influenced by chronic and acute inflammation status, so the body’s status can be well reflected by the results of a biochemical analysis of biomarkers. However, the identification of MetS is a long process that must fulfill all the requirements of the MetS definition. The present research subjects were adolescents in Beijing. In comparison with adults, young populations can avoid indirect impacts caused by the emergence of various chronic diseases or other physical disturbances. Therefore, we used MetS as the risk factor to illustrate the association between MetS and various biomarkers such as SUA, CRP, low density lipoprotein (LDL), and alanine aminotransferase (ALT) in the adolescents.

## 2. Materials and Methods

### 2.1. Study Design and Participants

A cross-sectional analysis was carried out, and data were extracted from the Nutrition and Health Surveillance in Primary and Secondary school students of Beijing (NHSPSB) 2017 study. Multistage stratified cluster sampling was performed to select participants. First, 3 urban districts and 4 suburban districts in Beijing were selected according to economic development levels and the working capability of the investigators. Second, 4 primary schools and 4 secondary schools were selected from each district via a simple random sampling method. Third, one class was chosen from the first, third, and fifth years of each primary school, respectively; only one class from the first year of each secondary school was selected. The criteria for class selection depended on the cooperation of the chief teacher, and only the most cooperative one was chosen at the end. All students and their parents (caregivers) from the selected classes gave informed consent before the students started to answer the questionnaires. All parents and selected chief teachers were trained by the investigator for completion of the questionnaires. Only questionnaires of first-year primary school students were filled in by their parents based on their observations, which were coordinated by an experienced investigator at the parents’ meeting. The rest of the students’ questionnaires were filled in by the students themselves under the guidance of the chief teacher within one hour. This study was approved by the institutional review board of the Beijing Center for Diseases Prevention and Control (201708).

Two different criteria were adopted for diagnosing MetS in children based on the Chinese criteria published in the Chinese Journal of Pediatrics [11]. The first one was set for children younger than 10 years old, and another one was set for children between 10 and 18 years old. However, it is inappropriate to give a diagnosis for children younger than 10 years old, as they are undergoing fast development in terms of physical growth. Thus, our present research only focused on adolescents aged 10–15 years old.

### 2.2. Anthropometric Measurement and Definition of Mets in Adolescents

All of the students’ heights and body weights were measured when they were barefoot and wearing light indoor clothes (height, SZG, Nantong Yuejian anthropometric equipment co., LTD, Nantong, China; weight, TC200K, Changshu C&G Measuring instrumental company, Changshu, China). Body mass index (BMI) was calculated using weight (kg) divided by the square of the height (m^2^). Blood pressure was measured using an electric sphygmomanometer (HBP-1300, Omron health care, Kyoto, Japan). Blood pressure was measured twice in the seated position after 5 min of rest by well-trained field investigators. A third blood pressure measurement was carried out if the first two blood pressure readings differed by more than 10 mmHg. The average blood pressure was calculated to determine the presence of hypertension. Waist circumference (WC) was measured at the midpoint between the iliac crest and the last rib via an inelastic tape (plastic tape measure, No. 8213, Deli stationery company, Ningbo, China), to the nearest 0.1 cm. Then, the waist-to-height ratio (WHtR) was calculated by dividing the WC by the height.

The standard applied for the assessment of overweight/obesity in adolescents was WS/T 586-2018 [12]. The criteria used to diagnose MetS in the present study were the fast screening criteria: (1) central obesity, measured by the WHtR was adopted in this study (≥ 0.46 for girls and ≥ 0.48 for boys); (2) impaired fast glucose was considered a fasting glucose concentration of ≥ 5.6 mmol/L; (3) hypertension, as indicated by a systolic blood pressure (SBP) of ≥130 mmHg or a diastolic blood pressure (DBP) of ≥ 85 mmHg; (4) a low concentration of HDL (< 1.03 mmol/L); and (5) a high concentration of TG (≥ 1.47 mmol/L). An individual was considered to have MetS if central obesity and at least two of the remaining four conditions were present.

### 2.3. Biochemical Analysis

Blood samples were obtained from participants who had fasted for at least 10 hours. These were stored in EDTA anti-coagulant tubes. SUA, TG, HDL, LDL, fasting blood glucose, aspartate transaminase (AST), ALT, homocysteine and CRP were measured in the CDC’s laboratory with a Hitachi (7600) automatic analyzer using Wako Diagnostics reagent, except for CRP (Roche, Basel, Switzerland), in accordance with the instructions of the manufacturer.

### 2.4. Statistical Analysis

All statistical analyses were carried out using Stata 14.0 software (StataCorp, College Station, TX, USA). Continuous variables were represented as the mean ± SD and categorical variables were represented as percentages. Student’s *t*-tests were applied for the comparison of continuous variables and χ^2^ was used for categorical variables. Multivariable linear regression was adopted for the association analysis. As most biochemical markers were skewed to one side rather than normally distributed, we took the logarithms of SUA, CRP, HDL, LDL, and ALT. After transformation, these indicators approached a normal distribution so that the statistical significance of coefficients of covariants could be directly tested in the multivariable linear regression. In model 1, only age (continuous) and sex (girl and boy) were adjusted for. In model 2, the annual household income per capita (<2000 Yuan; (2000–7000) Yuan; >70000 Yuan), the characteristics of the guardian such as education attainment (primary school, middle school, and college and above), sex (male and female), age (continuous), and place of residence (urban and suburban) were adjusted for as well. A two-sided *p*-value of <0.05 was considered statistically significant.

## 3. Results

### 3.1. General Characteristics of the Participants

A total of 1766 students (895 girls and 871 boys) were included in the present study, of which 669 came from urban areas and the other 1097 came from suburban areas. The mean (SD) height was 154.65 (10.56) cm for boys and 153.01 (8.72) cm for girls (Table 1). The average weight was 50.16 (14.95) kg for boys and 46.13 (12.36) kg for girls. The mean (SD) BMI was 20.64 (4.45) and 19.47 (3.92) for boys and girls, respectively. The prevalence of overweight was 17% in the total sample—21% in boys and 14% in girls. The prevalence of obesity was 20% in the total sample, 24% in boys, and 17% in girls. In conclusion, boys were taller, heavier, and had a higher risk of being overweight /obese in comparison with girls. Moreover, boys also had higher mean values of WHtR, HDL-C, fasting blood glucose, SBP, and DBP, but their TG concentration was lower than that of girls (0.80 vs. 0.88) (Table 2). The prevalence of MetS was 3% in the total sample—4% in boys and 2% in girls. Moreover, the prevalence of MetS in the overweight and obesity populations was 5% and 12%, respectively. The prevalence remained higher in boys than in girls. Interestingly, the prevalence of MetS in suburban areas was almost 2.5 times as large as the value in urban areas. Abdominal obesity tended to be the most important risk factor for MetS, and its prevalence was 30% in the total sample—34% in boys and 27% in girls. The second risk factor was impaired fasting glucose and its prevalence was 11% in the total sample—14% in boys and 8% in girls. The third risk factor was a high serum TG concentration, and a higher value was observed in girls. The fourth risk factor was elevated blood pressure, and its prevalence was 7% in the total sample—9% in boys and 5% in girls. The last one was a lower concentration of HDL. The prevalence was only 4% in the total sample—5% in boys and 4% in girls.

### 3.2. The Biochemical Indexes in Mets Children and Non-Mets Children

To further illustrate the health impacts of MetS, the concentrations of ALT, AST, cholesterol, SUA, HDL, homocysteine, LDL, and CRP were analyzed, as well (Table 3). The concentrations of all indicators were higher in the MetS children in comparison with non-MetS children, except for HDL. In addition, the differences in ALT, SUA, HDL, homocysteine, LDL, and CRP between the two groups of children were statistically significant (all *p* < 0.05). The values of ALT and CRP in MetS children were almost twice as high as those in non-MetS children (ALT: 23.45(2.7) VS. 12.85(0.25); CRP: 1.78(0.34) VS.0.98 (0.06)). The SUA concentration was 385.29 (13.25) *μ*mol/L for MetS children and 320.61(1.94) *μ*mol/L for non-MetS children. Although a significant difference was also detected for homocysteine, the magnitude was too small to be clinically significant. Therefore, it was not included in further analyses.

### 3.3. Association between MetS and Biochemical Indexes

A simple multivariable regression analysis indicated that MetS was positively associated with ALT, SUA, LDL, and CRP (Table 4). Those relationships still remained even after adjusting for co-variables in models 1 and 2. Of the five components, abdominal obesity and a high serum TG concentration were significantly associated with increased ALT, SUA, LDL, and CRP even after adjusting for co-variables in models 1 and 2. An elevated blood pressure was related to ALT, SUA, and CRP in the three models. However, a significant association between LDL and elevated blood pressure was only observed after adjusting for co-variables. Additionally, a lower HDL concentration was associated with higher UA and CRP concentrations, even after adjusting for co-variables, but there was no significant association with the concentration of LDL. Additionally, impaired fasting glucose was only associated with ALT and UA; however, after adjusting for co-variables, the association with UA became insignificant.

## 4. Discussion

Metabolic syndrome and its associations with ALT, SUA, LDL and CRP in Beijing children were depicted in the present study. Generally, the prevalence of MetS was higher in overweight and obese children and was higher in suburban areas. Metabolically, MetS was significantly associated with the ALT, SUA, LDL, and CRP concentrations. ALT was influenced by all five components and statistical significance remained with a certain decrease after adjusting with co-variables. SUA and CRP were associated with four components but not blood glucose. The statistical significance of the association with blood glucose disappeared after adjusting for age and sex, which indicates that blood glucose varied across adolescents of different ages and sexes. Interestingly, statistical associations between LDL and HDL/blood glucose were observed after adjusting for co-variables. Notably, a lower HDL concentration was not always accompanied by a higher LDL concentration since no significant association was detected between these two variables.

The present study found that the prevalence of MetS was 3%, which was similar to the urban prevalence (2.8%) detected by National surveillance [3]. We also found that boys had a higher prevalence rate. However, the results showed that the prevalence of MetS was higher in suburban areas of Beijing, contrary to the results derived from the National Surveillance. As one of the first-tier cities in China, the economic and culture development in Beijing are high in relation to the rest of China. It is plausible that residences in urban areas are aware of the risk of being obese and they might attain a higher educational level, so that they are more likely to maintain healthy lifestyles. In the present study, of the 5 components of MetS, impaired fasting glucose and a high serum TG concentration were the major common risk factors after abdominal obesity. This result is different from that shown by the national surveillance, which indicated that hypertension and a low HDL concentration are the major prevalent risk factors. The difference might be attributed to the different diagnostic criteria used. Intriguingly, our results were in accordance with the Cook’s criteria for Chinese children and adolescents, which also revealed that a high serum TG is the major risk factor [3]. Moreover, abdominal obesity, a high serum TG concentration, and impaired fasting glucose were found to be major prevalent risk factors for American children and adolescents when using Cook’s criteria. This phenomenon is consistent with our results.

Regarding the biochemical indexes, our results were in line with most research, revealing that MetS was related with SUA, ALT, and CRP. Plenty of documents have indicated the strong association between SUA and MetS. However, no paper has clearly illustrated the molecular mechanism of SUA and MetS yet [13]. Until now, there was no clear threshold to judge children’s SUA concentration. When we used the adult threshold for our present study (> 420 μmol/L), 16 out of 59 MetS students were classified as having hyperuricemia. Among the 16 students with hyperuricemia, 11 had high serum TG concentrations (≥ 1.47 mmol/L), 11 had high SBP values (≥ 130 mmHg) and 6 had impaired fasting glucose (≥ 5.6 mmol/L). However, all 16 hyperuricemia students had BMI values above 25, and 10 of them had BMI values higher than 28. Overweight and obesity were probably the underlying condition triggering the pathophysiologic abnormality [14]. Therefore, the present study still could not disentangle the causality between SUA and MetS.

Pediatric nonalcoholic fatty liver disease (NAFLD) is emerging as a comorbidity of MetS [15,16]. Inflammation and cellular injury are often considered as the biomarkers of NAFLD, particularly ALT [17]. Pediatric NAFLD was screened for via assessment for unexplained ALT elevation >30 u/L among adolescents [15]. The prevalence of an abnormal ALT level in the present study was 3.6%, which was lower than that shown for Hispanic and non-Hispanic adolescents. Additionally, it has been indicated that the prevalence of elevated ALT varies by ethnicity [18,19]. The prevalence of an abnormal ALT concentration was 22%, and the ALT concentration was tightly linked with all components of MetS. Previous documents indicated that an elevated ALT concentration is associated with waist circumference and insulin resistance, which is consistent with our results [20,21]. Of note, the cause–effect relationship between MetS and NAFLD showed some degree of overlap. Clearly, the presence of NAFLD is related to insulin resistance and inflammation [22,23]. The presence of NAFLD tended to be a predictor of MetS. Previous studies pointed out that NAFLD would be a reason for worsening MetS. Meanwhile, inflammation would drive abnormalities linked with MetS and NAFLD in the basic biological models [24], which might contribute to liver fat accumulation via related pathways [25].

CRP is an acute phase plasma protein which predicts the cardiovascular outcomes and alters glucose metabolism [26,27]. In the present study, the CRP level was significantly associated with MetS, especially several components such as central obesity, low HDL, and hypertension. This result is consistent with previous research [14]. CRP was mainly associated with obesity and hypertension. However, no significant correlation between CRP and impaired fasting glucose was detected, which might be attributed to the different study subjects used in our study compared with previous research [27]. Moreover, evidence suggests that CRP could be used to diagnose chronic inflammatory disease [28,29]. Currently, scientific inquiry into MetS and its underlying chronic inflammatory state has revealed that it had connections to several inflammatory dermatoses like psoriasis, hidradenitis suppurative, and atopic dermatitis [30,31,32]. Although these sequelae of MetS are not often seen until adulthood, researchers and caregivers should be aware of the potential comorbid disease burden among these pediatric patients. Additionally, LDL is strongly associated with central obesity, TG, and hypertension. Previous documents have provided evidence that LDL is a predictor of cardiovascular disease [33].

Several limitations of the present study should be mentioned. First, it was a cross-sectional study which cannot illustrate the causal relationship between MetS and other biochemical indexes. Second, the stage of puberty was not included as a co-variable in the present study due to data availability. Puberty could affect fat distribution and insulin sensitivity in the muscle and liver. Insulin sensitivity can decline by 20–50% during puberty and can return to normal at the end of pubertal development. Third, all participants lived in Beijing, which might restrict the extrapolation ability of the data.

Despite the aforementioned limitations, the present study is the first to reveal associations between MetS and SUA, ALT, LDL, and CRP in Chinese adolescent students with a moderate sample size.

## 5. Conclusions

In conclusion, our research indicates that the prevalence of MetS is high among Beijing adolescent students, and MetS has strong connections with ALT, SUA, and CRP. To further discover the causality between these biochemical markers, longitudinal or interventional studies should be carried out. In addition, the single risk factors of obesity, NAFLD and SUA were much higher among adolescent students. Early intervention strategies, such as dietary pattern intervention and physical excise, should be designed for that population to reduce the large disease burden of CVD, Type 2 diabetes, and steatohepatitis or even cirrhosis in adulthood.

## Figures and Tables

**Table 1 ijerph-16-04557-t001:** Characteristics of participants.

General Information	Boys (*n* = 871)	Girls (*n* = 895)	Total Population (*n* = 1766)
**Educational attainment of caregivers**			
Primary school and below	21 (2%)	22 (2%)	43 (2%)
Middle school	398 (46%)	397 (44%)	795 (45%)
College and above	452 (52%)	476 (53%)	928 (53%)
**Annual household income per capita (Yuan)**			
< 20000	255 (29%)	289 (32%)	544 (31%)
20000–70000	376 (43%)	358 (40%)	734 (42%)
> 70000	240 (28%)	248 (28%)	488 (28%)
**Residence**			
Urban	539 (62%)	558 (62%)	1097 (62%)
Suburban	332 (38%)	337 (38%)	669 (38%)
Age of participants (Years)	11.34 ± 1.12	11.26 ± 1.13	11.30 ± 1.12
Height (cm)	154.65 ± 10.56	153.01 ± 8.72	153.82 ± 9.70
Weight (kg)	50.16 ± 14.95	46.13 ± 12.36	48.12 ± 13.84
Waist circumference (cm)	70.77 ± 11.99	66.02 ± 9.65	68.36 ± 11.12
Body Mass Index (BMI)	20.64 ± 4.45	19.47 ± 3.92	20.05 ± 4.23
Overweight	184 (21%)	122 (14%)	306 (17%)
Obesity	209 (24%)	148 (17%)	357 (20%)

**Table 2 ijerph-16-04557-t002:** Distribution of risk factors of MetS among participants.

Risk Factors of Mets	Boys (*n* = 871)	Girls (*n* = 895)	Total Population (*n* = 1776)
Waist to height ratio (WHtR)	0.46 ± 0.07	0.43 ± 0.06	0.44 ± 0.06
Serum Triglycerides (mmol/L)	0.80 ± 0.51	0.88 ± 0.50	0.84 ± 0.51
High Density Lipoprotein (HDL) (mmol/L)	1.54 ± 0.34	1.53 ± 0.32	1.54 ± 0.33
Fasting blood glucose (mmol/L)	5.19 ± 0.43	5.08 ± 0.37	5.14 ± 0.41
Systolic blood pressure (mmHg)	113.28 ± 11.57	110.98 ± 10.73	112.11 ± 11.21
Diastolic blood pressure (mmHg)	64.53 ± 7.83	65.60 ± 7.57	65.07 ± 7.71
MetS (%)	38 (4%)	21 (2%)	59 (3%)
Normal weight MetS (%)	1 (0.2%)	1 (0.2%)	2 (0.2%)
Overweight MetS (%)	11 (6%)	3 (3%)	14 (5%)
Obesity MetS (%)	26 (12%)	17 (12%)	43 (12%)
Urban MetS (%)	6 (2%)	6 (2%)	12 (2%)
Suburban MetS (%)	32 (6%)	15 (3%)	47 (4%)
**MetS components (%)**			
Abdominal adiposity	296 (34%)	242 (27%)	538 (30%)
High serum triglycerides (TG)	78 (9%)	93 (10%)	171 (10%)
Low HDL	42 (5%)	36 (4%)	78 (4%)
Abnormal glucose homeostasis	118 (14%)	69 (8%)	187 (11%)
Evaluated blood pressure	80 (9%)	41 (5%)	121 (7%)

**Table 3 ijerph-16-04557-t003:** Concentrations of biochemical markers in non-MetS and MetS students.

Biochemical Markers	MetS Students (59)	Non-MetS Students (1707)	*p*-Value
ALT (U/L)	23.45 ± 2.70	12.85 ± 0.25	<0.01
AST (U/L)	23.04 ± 1.37	22.07 ± 0.16	0.29
Cholesterol (mmol/L)	4.2 ± 0.10	4.14 ± 0.02	0.73
SUA (umol/L)	385.29 ± 13.25	320.61 ± 1.94	<0.01
HDL (mmol/L)	1.09 ± 0.19	1.55 ± 0.32	<0.01
Homocysteine(umol/L)	14.78 ± 0.79	13.38 ± 0.14	0.07
LDL (mmol/L)	2.63 ± 0.08	2.28 ± 0.01	<0.01
CRP (mg/L)	1.78 ± 0.34	0.98 ± 0.06	0.01

The Student’s *t*-test was applied to test the mean equality between these two groups. ALT: alanine aminotransferase; AST: aspartate transaminase; SUA: serum urine acid; HDL: high density lipoprotein; LDL: low density lipoprotein; CRP: C-reactive protein.

**Table 4 ijerph-16-04557-t004:** Association between MetS and biochemical markers (in logarithm values).

	Unadjusted		Model 1		Model 2	
	Coef *	*p*	Coef	*p*	Coef	*p*
MetS						
ALT	0.48	<0.01	0.45	<0.01	0.45	<0.01
SUA	0.18	<0.01	0.14	<0.01	0.15	<0.01
LDL	0.14	<0.01	0.15	<0.01	0.18	<0.01
CRP	0.91	<0.01	0.86	<0.01	0.86	<0.01
Abdominal adiposity						
ALT	0.35	<0.01	0.34	<0.01	0.33	<0.01
SUA	0.15	<0.01	0.15	<0.01	0.15	<0.01
LDL	0.11	<0.01	0.10	<0.01	0.11	<0.01
CRP	0.93	<0.01	0.90	<0.01	0.91	<0.01
High serum TG						
ALT	0.27	<0.01	0.28	<0.01	0.28	<0.01
SUA	0.11	<0.01	0.11	<0.01	0.11	<0.01
LDL	0.16	<0.01	0.16	<0.01	0.17	<0.01
CRP	0.40	<0.01	0.43	<0.01	0.42	<0.01
Low HDL						
ALT	0.12	0.02	0.12	0.02	0.11	0.04
SUA	0.08	<0.01	0.06	0.03	0.06	0.02
LDL	–0.03	0.29	–0.02	0.46	0.00	0.94
CRP	0.71	<0.01	0.70	<0.01	0.70	<0.01
Abnormal glucose homeostasis						
ALT	0.11	<0.01	0.08	0.02	0.08	0.02
SUA	0.05	<0.01	0.01	0.55	0.01	0.52
LDL	0.01	0.64	0.02	0.26	0.03	0.10
CRP	0.07	0.42	0.02	0.84	0.02	0.79
Evaluated blood pressure						
ALT	0.29	<0.01	0.26	<0.01	0.26	<0.01
SUA	0.17	<0.01	0.13	<0.01	0.13	<0.01
LDL	0.05	0.06	0.06	0.02	0.07	0.01
CRP	0.48	<0.01	0.42	<0.01	0.42	<0.01

Biochemical markers are measured in logarithm so that they are more likely to follow a normal distribution. ALT: alanine aminotransferase; SUA: serum urine acid; HDL: high density lipoprotein; LDL: low density lipoprotein; CRP: C-reactive protein. Model 1 adjusted for age and sex of participants. Model 2 adjusted for age and sex of participants individually, annual household income per capital, residents’ localization, and characteristics of participants’ caregivers (educational attainment, age, and sex). *: “Coef” indicates coefficient of MetS and its components.

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
