# Peer review of "The Association between Metabolic Syndrome and Biochemical Markers in Beijing Adolescents"

_ijerph, 2019, doi:10.3390/ijerph16224557_

Round 1
Reviewer 1 Report
Comment to the Authors
In this paper, the authors report the prevalence of MetS in adolescents in Beijing and its association with the concentration of alanine aminotransferase, serum uric acid, low density lipoprotein, and C-reactive protein. The study is well conducted, even if it does not appear to generate significant conceptual advance beyond published literature.
The main concern regards the statistical study. In fact, I think that continuos variables have a non-normal distribution and, then, non parametric test must be employed. Also, in linear regression analysis (tab 4) it has been indicate a “Coef”: what is this?
There are several typo errors.
Author Response
Reply: Thanks a lot for your valuable comments. We agree that this paper might not indicate advance idea in comparison with current literature. However, using Chinese adolescents to verify the published results is meaningful as well. At least Chinese adolescent do have metabolic disorders in the presence of obesity/metabolic syndrome. Our study is a preliminary research, which provokes some new ideas and more comprehensive studies should be carried out in the future.
Regarding the statistical analysis, we tested the distribution of continuous variables (biochemical markers) and did find that they have a non-normal distribution. We thus take logarithm of all biomarkers. After transformation, all indicators are approximately normal-distribution. We run all regressions again and present new results in the revised manuscript. In general, the main results are the same as the old one.
The “coef.” refers to the coefficient of MetS and its components. In the new results, it can be explained as the risk difference between people with MetS (or its components) and those without MetS (or its components). Thank you very much for figuring out this mistake.
Reviewer 2 Report
Dear editors:
Yao Zhao and the co-authors have evaluated the association between metabolic syndrome and uric acid, ALT, LDL and CRP in adolescents in Beijing. I have a number of major problems concerning this study:
From the perspectives of epidemiologic studies, the sample size is too small to present the current prevalence of metabolic syndrome in cross-sectional survey in Beijing adolescents. For example, Gu D et al reported a nationally representative sample of 15,540 Chinese adults [Prevalence of the metabolic syndrome and overweight among adults in China. Lancet, 2005. 365(9468): p. 1398-405]. Epidemiology studies aim to investigate the distribution of diseases and other health-related conditions in populations. Investigators applied such studies to control health problems via prevention. Thus the purpose of epidemiology is to understand what risk factors are associated with a specific disease. The study interest is focused on the novel and unknown risk factors. However, it is well known that all of the parameters (uric acid, ALT, NAFLD, LDL and CRP) are robustly associated with metabolic syndrome. Nothing new in this manuscript limits the scientific merits. Extensive editing of English language and style were required. The term of IFG “impaired fast glucose” should be rephrased as “impaired fasting glucose”. The abbreviation of “serum uric acid (SUA)” should be rephrased as “uric acid (UA)”, and the term of “Serum urine acid” is incorrect.Author Response
Reply: Thanks a lot for your careful and valuable comments. We agree that large sample size is important for a representative epidemiological research. We have tried our best to collect as most students as we could under the budget constraint. The 1766 students were selected from 56 primary and secondary schools in 7 districts in Beijing using a multistage stratified cluster sampling strategy. So that our sample can still be taken as a representative sample of the 500 thousand adolescents aged between 10-15 years old in Beijing.
The contribution of our research is that we not only reveled the associations between MetS and SUA, ALT, LDL and CRP in Chinese adolescent students, but also investigated the associations between MetS components (ALT, AST, cholesterol, SUA, homocysteine, LDL and CRP) and the above mentioned biomarkers (comprehensively ALT, SUA, LDL, CRP).
Regarding the English language and style, we have invited a native speaker to polish our writing. Further language editing service will employed upon request.
The term IFG “impaired fast glucose” has been changed to “impaired fasting glucose”. Thank you very much for your suggestion. However, we insist on using SUA as the abbreviation of serum uric acid after reading several published papers (listed below). We think it is better to indicate the uric acid we tested is in the blood, because uric acid can be detected in urinary as well.
References:
Dai X, Yuan J, Yao P et.al., Association between serum uric acid and the metabolic syndrome among a middle- and old-age Chinese population. Eur J Epidemiol.2013 Aug;28(8):669-76 Voruganti VS, Laston S, Haack K et.al.Serum uric acid concentration and SLC2A9 genetic variation in Hispanic Children: the Viva La Familia Study. Am J Clin Nutr.2015 Apr;101(4):725-32 Yang T, Chu CH, Bai CH et.al., Uric acid level as a risk marker for metabolic syndrome: A Chinese cohort study. Atherosclerosis.2012 Feb;220(2):525-31Reviewer 3 Report
In the present study, it was chosen as a variable to belong to an urban or suburban environment and the family´s income. However one of the most important determinants in childhood obesity is the cultural level of the parents.
Has this variable been taken into account?
Thank you
Author Response
Responses: Thanks a lot for your kind reminding. We totally agree that parents’ cultural level plays an important role in children’ obesity. We thus adopted the education attainment of parents (or care givers)_to measure the cultural level of the parents, which was classified into three levels: primary, middle schools, and college and above. It was adjusted in model 2 (Table 4).
Round 2
Reviewer 2 Report
Dear editors:
Yao Zhao and the co-authors have refined the article to address the association between metabolic syndrome and uric acid, ALT, LDL and CRP in adolescents in Beijing. I have several minor concerns:
Beijing, the capital of China, is one of the famous cities in the world. Thus the epidemiological data of Beijing adolescents is of prime importance. In my opinion, the results about parameters of socioeconomic status could also be the selling point. Socioeconomic status encompasses not just income but also educational attainment and social class. Socioeconomic status can influence health status as well as quality of life among people within society. The title could be rephrased as “The association between metabolic syndrome, biochemical markers and socioeconomic status in Beijing adolescents.” The biomarkers in the study were not limited to “uric acid, ALT, LDL and CRP”. Line 39: insert the term “Socioeconomic status”. Line 45: while high density lipoprotein (HDL) was lower in MetS children=> A dot is needed. Line 46: Please add “After adjusting for socioeconomic parameters in the multivariable regression model, MetS was strongly associated with ALT, SUA, HDLand LDL.” A space should be inserted in “HDLand”. Conclusion could be rephrased as “The epidemiological issues of MetS in Beijing adolescents should be aware across socioeconomic class.” Abstract: Objective: metabolic related biochemical markers=> MetS related biochemical markers. A gentle reminder: The scientific merit of the current study might be limited due to frequent mistakes in the manuscript. Extensive editing of English language and style were still required. The MDPI English Editing Service is suggested.Author Response
Reply: Thanks a lot for your kind advice. We have changed the title of our study and corrected all the typos. We send our paper for English editing as well. We do found the language of our paper needs to be improved for better understanding. Thanks again for your kind suggestion.
